# Role of Hybrid Nano-Zinc Oxide and Cellulose Nanocrystals on the Mechanical, Thermal, and Flammability Properties of Poly (Lactic Acid) Polymer

**Dilpreet S Bajwa [1,*], Jamileh Shojaeiarani [2], Joshua D. Liaw [3] and Sreekala G Bajwa [4]**

[1] Department of Mechanical and Industrial Engineering, Montana State University, Bozeman, MT 59717, USA

[2] Department of Mechanical Engineering, Western New England University, Springfield, MA 01119, USA; jamileh.shojaeiarani@wne.edu

[3] Department of Mechanical Engineering, North Dakota State University, Fargo, ND 58102, USA; joshualiawdx@hotmail.com

[4] College of Agriculture, Montana State University, Bozeman, MT 59717, USA; sreekala.bajwa@montana.edu

\* Correspondence: dilpreet.bajwa@montana.edu

**Abstract:** Biopolymers with universal accessibility and inherent biodegradability can offer an appealing sustainable platform to supersede petroleum-based polymers. In this research, a hybrid system derived from cellulose nanocrystals (CNCs) and zinc oxide (ZnO) nanoparticles was added into poly (lactic acid) (PLA) to improve its mechanical, thermal, and flame resistance properties. The ZnO-overlaid CNCs were prepared via the solvent casting method and added to PLA through the melt-blending extrusion process. The composite properties were evaluated using SEM, a dynamic mechanical analyzer (DMA), FTIR TGA, and horizontal burning tests. The results demonstrated that the incorporation of 1.5% nano-CNC-overlaid ZnO nanoparticles into PLA enhanced the mechanical and thermal characteristics and the flame resistance of the PLA matrix. Oxidative combustion of CNC-ZnO promoted char formation and flame reduction. The shielding effect from the ZnO-CNC blend served as an insulator and resulted in noncontinuous burning, which increased the fire retardancy of nanocomposites. By contrast, the addition of ZnO into PLA accelerated the polymer degradation at higher temperature and shifted the maximum degradation to lower temperature in comparison with pure PLA. For PLA composites reinforced by ZnO, the storage modulus decreased with ZnO content possibly due to the scissoring effect of ZnO in the PLA matrix, which resulted in lower molecular weight.

**Keywords:** polylactic acid; cellulose nanocrystals; nano zinc oxide; physical and mechanical properties

## 1. Introduction

The global concern over reducing the environmental impact caused by petroleum-based polymers has accelerated research aimed at developing biopolymer materials from sustainable resources [1]. Biopolymers with universal accessibility and inherent biodegradability can offer an appealing sustainable platform to supersede petroleum-based polymers. Among different types of biopolymers, poly(lactic acid) (PLA) as the most commercially available biopolymer is a reliable alternative for petroleum-based polymers in different industries. PLA is used in diverse applications by different industries ranging from medical devices, packaging, and consumer products to automobile composites due to its superior transparency, easy processability, and high mechanical properties [2,3]. PLA-based nanocomposites are known for improved properties such as stiffness, thermal stability, biodegradability, and lower permeability [4]. However, the inherent limitations

of PLA such as low thermal properties and high flammability need to be addressed to extend the potential applications of PLA [5,6].

The incorporation of different types of nanofillers as fire retardants is the most common method for improving the fire retardancy of polymers, owing to the ease of fabrication and high efficiency [7]. Zinc oxide (ZnO) nanoparticles are relatively nontoxic and environmentally friendly multifunctional inorganic fillers, which are certified as generally recognized as safe materials (GRAS). Lately, polymer composites with nano-ZnO particles have gained considerable attention due to their low refractive index, high optical transparency, flame retardancy, and nontoxicity [8]. ZnO nanoparticles can be added to various polymers to improve different polymer properties such as thermal stability, stiffness, fire retardancy, and permeability [4]. To endow ZnO with exceptional properties, some researchers suggested the surface modification of ZnO or the incorporation of ZnO into a polymer in the form of nanohybrids [9]. Contradictory observations have been reported regarding the incorporation of ZnO into a PLA matrix. It was reported that ZnO nanoparticles can act as a disruptor in a PLA matrix, deteriorating molecular weights, and thermal and rheological properties of composites [10].

ZnO nanocrystals tend to aggregate through Ostwald ripening due to their high surface energy [11]. To overcome this weakness, hybrid cellulose nanocrystal-ZnO nanohybrids have been synthesized and evaluated owing to the electrostatic interactions between cellulose nanocrystals (CNCs) and ZnO, which can improve the dispersibility and bring new functions to the host polymer. These nano-hybrid mixtures have been applied for controlling photolytic activity, UV protection, and for antibacterial performance [8,11,12]. Awan et al. have summarized the morphology of hybrid CNC-ZnO structures such as leaf-like growth, flower-like nanorod clusters, spheres, sheet-like irregular discs, and hexagonal wurtzite [11]. Sheet-like CNC-ZnO nanohybrids developed using the one-step hydrothermal process are reported to act as multifunctional reinforcing agents, a UV absorber, and antibacterial agents in biopolyester (poly-3-hydroxybutyrate-co-3-hydroxy valerate, PHBV). The hybridization of cellulose nanocrystals (CNCs) and nano-ZnO resulted in high thermal stability in the composition [13]. This behavior was attributed to the strong interaction between oxygen atoms of the CNCs and ZnO, suggesting the formation of a protective barrier for CNCs against thermal decomposition. Cellulose nanocrystals (CNCs) with nanoscale dimension, high aspect ratio, high surface area, and highly crystalline structure have attracted a great deal of attention as reinforcing fillers to improve the mechanical and thermal properties of polymers [1]. CNCs are hydrophilic in nature, meaning they can be uniformly dispersed in hydrophilic matrices; however, due to the hydrophobic nature of PLA, agglomerates are formed when CNCs are incorporated into PLA, due to lack of compatibility between the hydrophilic CNCs and the hydrophobic PLA [6]. The homogenous dispersion of CNCs within a polymer matrix is key to benefit from their astounding properties [13]. It was previously reported that the incorporation of CNCs in PLA through chemistry-oriented surface modification treatment and the application of masterbatch followed by melt blending can result in the superior dispersion of CNCs in the PLA matrix [14].

Understanding the combined effects of nanosized inorganic metal oxides and cellulose crystals in biodegradable polymer (PLA) provides a new approach for improving the physico-mechanical properties using nanotechnology. In this context, a systematic investigation of thermal stability, dynamic mechanical properties, and flammability characteristics of CNCs/ZnO nanohybrids-filled PLA was conducted using multiple complementary techniques. CNCs were modified by the adsorption of nano-ZnO particles on the surface not only to improve the dispersion in the PLA matrix but also to increase the thermal properties of the host polymer. The underlying hypothesis was that the CNC would reinforce the polymer matrix and the CNC-ZnO complex would create a char/barrier inhibiting flame spread. Overall, the research is aimed at developing and evaluating a safe, effective, and environmentally friendly fire-retardant system (Figure 1). Furthermore, we

believe that the hybrid nano-ZnO and CNC blend in PLA composites would provide multifunctional properties such as fire resistance, UV stability, anti-bacterial effects, and superior optical transparency for use in commercial applications.

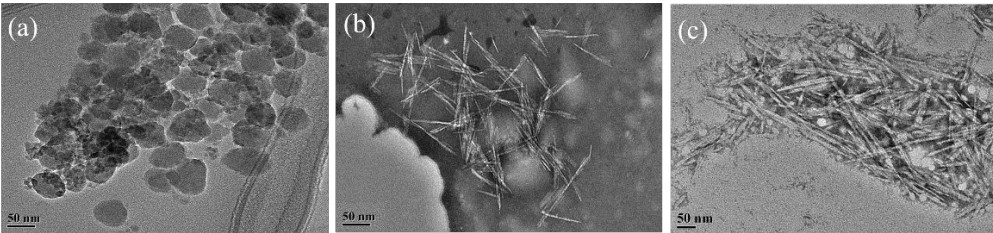

**Figure 1.** The morphology of ZnO-overlaid cellulose nanocrystals (CNCs) made from ZnO nanoparticles and CNCs: (**a**) ZnO, (**b**) CNCs, and (**c**) CNCs-ZnO.

## 2. Materials and Methods

### 2.1. Materials

Poly (lactic acid)-grade Ingeo™ 2003D manufactured by NatureWorks® (Minnetonka, MN) was used as a polymer matrix resin. The resin specifications were density (1.25 g/cm$^3$), melt flow rate (6 g/10 min), glass transition temperature (55–60 °C), and melting temperature (145–160 °C). Cellulose nanocrystals (CNCs) with dimensions of 10–15 nm width and 80–100 nm length were kindly donated by USDA Forest Products Laboratory (Madison, WI, USA). They were extracted from softwood pulp using the sulfuric acid hydrolysis process. The CNC was desulfated using hydrothermal treatment. Zinc acetate dihydrate with a molecular weight of 219.51 g/mol and a purity of 99.99% was purchased from Sigma-Aldrich (St. Louis, MO, USA). Sodium hydroxide with a molecular weight of 40.00 g/mol was obtained from BDH Chemicals (Radnor, PA, USA). Chloroform and methanol were purchased from Fischer Scientific (Waltham, MA, USA).

### 2.2. Synthesis Procedure of Zinc Oxide Nanoparticles

The ZnO nanoparticles were prepared using the procedure reported in the previous studies [15]. The first step involved preparing solution A: 0.02 mol of zinc acetate dihydrate pellets were added to 50 mL of methanol, and the solution was heated to 50 °C with continuous stirring for 30 min. The second step created solution B: 0.04 mol of sodium hydroxide pellets were added in 50 mL of methanol, and the solution was heated to 50 °C with continuous stirring for 60 min. To make the ZnO nano-sol, solution B was added dropwise to solution A under continuous stirring for 30 min. Then, the stirring was stopped, and the nano-sol was heated to 50 °C for 30 min. The nano-sol was again stirred for another 2 h without heat to create a transparent white sol gel. Finally, the white sol gel was placed in the oven at 80 °C until all the solvent evaporated.

### 2.3. Masterbatch Preparation

PLA/CNC/ZnO masterbatch thin films were prepared using the solvent casting method. The CNC and ZnO nanoparticles were thoroughly blended to achieve a uniform blend of ZnO-overlaid CNCs. This was conducted in a chloroform solution by the application of homogenization and ultrasonication to achieve a uniform dispersion. The concentration of PLA in chloroform was 7 wt.%. An ice bath was used during the process of homogenization and ultrasonication to prevent overheating of the samples. PLA pellets were dissolved in the chloroform at 50 °C for 24 h to create a solution. Then, the ZnO-CNCs suspension was added to the PLA solution and the mixture was homogenized in an ice bath for 5 min to ensure that the CNC and ZnO were well-dispersed in the PLA solution. The percentage of each component in the PLA-CNC-ZnO masterbatch is shown in Table 1. At the end, the mixture was poured into petri dishes, and to minimize the formation of air bubbles, mixtures were stored in the closed cabinet for 24–48 h until the

films were completely solidified. A secondary agglomeration of CNC-ZnO upon drying was observed on the nanocomposite sample under SEM microscopy analysis. The secondary agglomeration in the PLA matrix could be due to the unstable dispersion of CNC-ZnO in the PLA solution.

### 2.4. Nanocomposite Samples Preparation

The masterbatch films were chopped into small pieces to the approximate size of PLA pellets using a paper cutter. Prior to melt processing, PLA pellets and masterbatch pieces were oven-dried at 50 °C for 8 h to remove excess moisture from materials. The nanocomposite sheets were produced by diluting the masterbatch pellets through the extrusion process, as shown on Table 1. A twin-screw extruder (Leistritz Mic18/Gl-40D) equipped with a rectangular die was used to form the nanocomposite sheets with dimensions of 137 mm × 12 mm × 5 mm. To prevent the nanofiller thermal degradation, the processing temperature was maintained between 150 °C and 160 °C and the RPM was set at 150 rpm.

### 2.5. Scanning Electron Microscopy (SEM)

The morphology of the fractured surface and dispersion of CNC and nano-zinc in the PLA matrix were analyzed using SEM. Cross-sectional images were taken at three magnifications using a field emission scanning electron microscope (Supra 55VP, Zeiss, Thornwood, NY, USA) in the Image and Chemical Analysis Laboratory at Montana State University. The sample was coated with a thin film of Ir for 30 s for conductivity.

### 2.6. Transmission Electron Microscopy (TEM)

The structure and dimension of CNCs and ZnO nanoparticles, and the CNCs-ZnO nanohybrid were studied using transmission electron microscopy (TEM, JEM-2100, Peabody, MA, USA) operating at 2 kV. The aqueous suspensions of CNCs, ZnO, and CNCs-ZnO were stained by phosphotungstic acid prior to imaging to prevent charging and enhance the image contrast. The dimension of nanoparticles was measured using online available image processing software (Image J).

**Table 1.** Formulation of masterbatch and composite samples.

| Sample | Masterbatch | | | Extruded Sheets | | |
|---|---|---|---|---|---|---|
| | PLA (%) | CNCs (%) | ZnO (%) | PLA (%) | CNCs (%) | ZnO (%) |
| PLA | 100 | 0 | 0 | 100 | 0 | 0 |
| 1.5CNC | 90 | 10 | 0 | 98.50 | 1.50 | 0 |
| 1.5ZnO | 90 | 0 | 10 | 98.50 | 0 | 1.50 |
| 1.5CNC/1.5ZnO | 80 | 10 | 10 | 97.00 | 1.50 | 1.50 |
| 1.5CNC/2.5ZnO | 85 | 5 | 10 | 96.00 | 1.5 | 2.50 |

### 2.7. Dynamic Mechanical Analysis (DMA)

A TA Instruments Dynamic Mechanical Analyzer (DMA Q800, New Castle, USA) was used to understand the dynamics storage and loss modulus of the samples. The viscoelastic properties of samples were also studied through the tanδ curve. Selected samples with dimensions of approximately 59 mm × 12 mm × 5 mm were tested using the dual cantilever bending mode at a constant frequency of 1 Hz. The test was performed in the temperature range of 25 °C to 110 °C and the heating rate was set at 5.00 °C/min. For each formulation, three replicates were tested, and the mean values are reported.

### 2.8. Fourier-Transform Infrared Spectroscopy (FTIR)

FTIR was conducted using a Thermo Scientific Nicolet 8700 spectrometer in photoacoustic mode in the range of 400–3500 cm$^{-1}$. The samples were the small pieces having an approximate thickness of 0.3–0.5 mm. The number of scans was 32, with a wavenumber resolution of 4 cm$^{-1}$.

### 2.9. Thermogravimetric Analyses (TGA)

TGA measurements were carried out using a TA Instruments Thermogravimetric Analyzer (DSC Q500, New Castle, USA) on samples of about 7–8 mg. At least three replicates were scanned over a temperature range from 25 °C to 600 °C at a heating rate of 10 °C/min under a flowing nitrogen atmosphere with a flow rate 20 mL/min. One set of samples was tested under thermo-oxidative degradation mode under 70% oxygen and 30% nitrogen atmosphere.

### 2.10. Melt Flow Index

The melt flow index of the extruded pellets was determined using a melt flow instrument MP600 Extrusion Plastometer (Tinius Olsen Inc., Horsham, Pa.). The melt was extruded with a weight of 2.16 kg and the temperature inside the bore of the cylinder was set as 190 °C. The melt index was measured according to ASTM D1238.

### 2.11. Horizontal Burn Test

The flame retardance properties of samples were characterized using the horizontal burn test. The horizontal burn test was conducted on a rectangular bar specimen with dimensions of 135 mm × 12 mm × 5 mm in accordance with the ASTM D635–14 standard. The samples were fixed to the mount and its longitudinal axis was kept in the horizontal way during the test. The Bunsen burner was tilted and the flame was applied to the sample free end at 45°. The linear burning rate and type of burning were reported. Three specimens were tested for each formulation, and the data reported are the average values of the measurements.

## 3. Results and Discussion

### 3.1. Transmission Electron Microscopy (TEM)

The morphology of ZnO nanoparticles and CNCs-ZnO were studied using TEM images as shown in Figure 2. Fine spherical nanoparticles in the range of 15–65 nm can be observed for ZnO nanoparticles (Figure 2a), and CNCs are needle-like elongated nanoparticles with average widths ranging from 8 to 12 nm and lengths of 100–150 nm. Figure 2b shows spherical ZnO nanoparticles with a narrow size relatively well dispersed within rod-shaped CNCs nanoparticles.

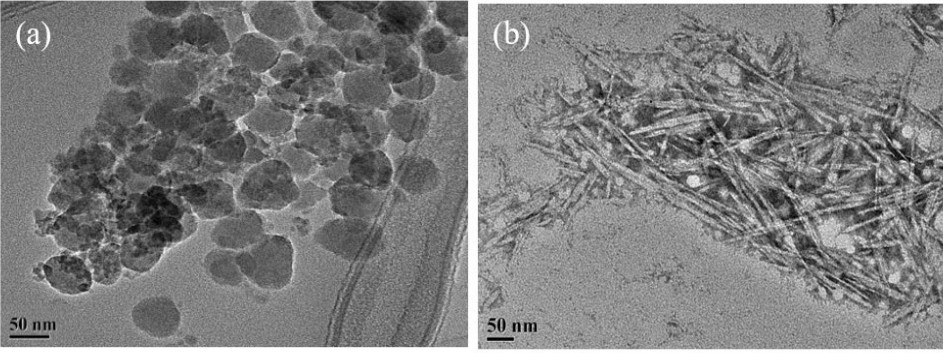

**Figure 2.** Transmission electron microscopy micrographs of a dilute suspension of (**a**) nano-ZnO, and (**b**) CNCs-ZnO particles.

### 3.2. Fourier-Transform Infrared Spectroscopy (FTIR)

FTIR spectra for the PLA, CNC, and composite containing nano-zinc oxide are shown in Figure 3. The characteristic PLA bands are observed at 868 and 1081 cm$^{-1}$ representing the flexural C-H bond and alkyl-ketone chain vibrations, respectively. The fast cooling of PLA during processing resulted in a slight shift in the lower band due to reduced crystallization, and this observation agrees with a previous reported work [16]. For CNC, the peak absorption bands at 3340 and 1380 cm$^{-1}$ reflect stretching and bending vibrations of hydroxyl groups, respectively, and peaks at around 1068 cm$^{-1}$ can be linked to the C–O–C stretching of pyranose and glucose ring skeletal vibration. In the PLA-CNC-ZnO composites, the nano-ZnO peak at 445 cm$^{-1}$ is due to Zn–O stretching vibration, which also confirms the nanocrystalline character of the resulting ZnO nanoparticles in the nanohybrids. In general, slightly higher peaks were observed for all three individual components of the composites, and this can be related to the strong interaction between oxygen groups and ZnO nanoparticles, the hydroxyl group of CNC, and the effect of cooling on the polymer matrix [17].

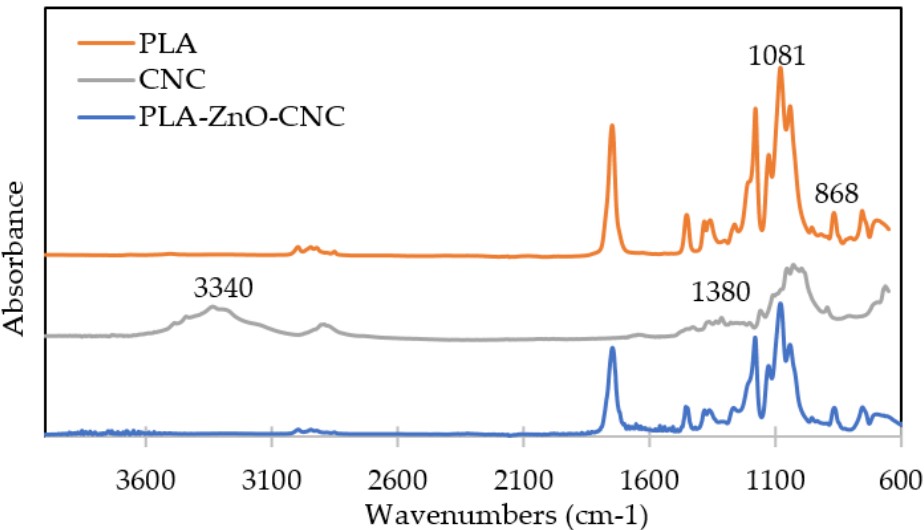

**Figure 3.** Infrared spectra of polylactic acid, cellulose nanocrystals, and a composite with nano-zinc oxide.

### 3.3. Dynamic Mechanical Analysis (DMA)

Dynamic viscoelastic properties such as the storage modulus (E′), loss modulus (E″), glass transition temperature ($T_g$), and loss factor (damping parameter, tanδ) were studied through dynamic mechanical analysis as shown in Table 2.). The variation in storage modulus of pure PLA and the corresponding composite as a function of composition and temperature while the samples were subjected to an elevated temperature with a constant rate is shown in Figure 4. Three regions (glassy, transition, and rubbery) were observed for all samples, and a sudden drop in the storage modulus showed the occurrence of the glass transition region of around 75–80 °C for all samples. The glass transition temperature was determined at the point at which a material changes from the glassy region to their rubbery state.

**Table 2.** Mechanical properties of each formulation from dynamic mechanical analysis (DMA).

| Sample | Peak tanδ | $T_g$ (°C) | Storage Modulus (MPa) | | Loss Modulus (MPa) | |
|---|---|---|---|---|---|---|
| | | | 35 °C | 70 °C | 35 °C | 70 °C |
| PLA | 1. 8 ± 0.6 | 79.1 ± 2.75 | 2384.0 ± 214.0 | 13.3 ± 2.1 | 40.7 ± 0.5 | 481.7 ± 0.7 |
| 1.5CNC | 1. 5 ± 0.3 | 80.6 ± 4.8 | 2642.3 ± 562.1 | 17.6 ± 0.9 | 34.5 ± 1.2 | 23.6 ± 1.5 |
| 1.5ZnO | 1.6 ± 0.3 | 79.6 ± 6.1 | 2014.6 ± 458.7 | 9.3 ± 2.4 | 58.2 ± 1.6 | 210.8 ± 2.2 |
| 1.5CNC/1.5ZnO | 1.6 ± 0.3 | 79.0 ± 4.3 | 2226.3 ± 310.9 | 6.1 ± 2.5 | 33.3 ± 2.3 | 52.4 ± 2.7 |
| 1.5CNC/2.5ZnO | 1.7 ± 0.3 | 78.9 ± 5.5 | 1587.2 ± 332.5 | 5.7 ± 1.9 | 30.9 ± 2.1 | 144.6 ± 3.1 |

The addition of solely cellulose nanocrystals led to a significant increase in storage modulus in nanocomposites as compared to pure PLA, which is attributed to the stiffing effect of CNCs. However, in nanocomposites reinforced by ZnO or the ZnO-CNC blend, the storage modulus decreased with the increase in ZnO concentration from 1.5% to 2.5%. Over the temperature range in this study, the formulation with 1.5% CNCs exhibited the highest storage modulus value, while that with 1.5% CNCs and 2.5% ZnO had the lowest. This observation could possibly be due to the scissoring effect of ZnO in the PLA matrix, which resulted in lower molecular weight and lower storage modulus, as observed in our previous study [10]. Furthermore, some previous studies have reported that the addition of ZnO leads to the intensive degradation of PLA chains at a high temperature [4,10]. In fact, ZnO serves as a reactant and accelerator in the degradation reaction of PLA and ac-

celerates the degradation of PLA at higher concentrations. In which Zn attacks the carbonyl group of PLA, resulting in chain scission and formation of a Zn carboxylic salt, it reacts with acidic compounds in PLA [18].

As it can be observed, the samples with 1.5% ZnO exhibited higher storage moduli than 1.5CNCs/2.5ZnO reinforced by 2.5% ZnO. The higher concentration of ZnO provokes the CNCs' intensity to absorb ZnO and this, in turn, increased the possibility of the formation of relatively large and unstable agglomerates in the polymer matrix. Furthermore, the increased concentration of ZnO led to the degradation of PLA, as explained earlier. However, the addition of CNCs compensated for the decline in the storage modulus in nanocomposites with the same ZnO content (1.5 ZnO and 1.5 CNC/1.5ZnO). This observation can be attributed to the interaction between ZnO and CNCs resulting from the electrostatic interaction between the oxygen atoms of hydroxyl groups on the surface of CNCs and $Zn^{2+}$ ions.

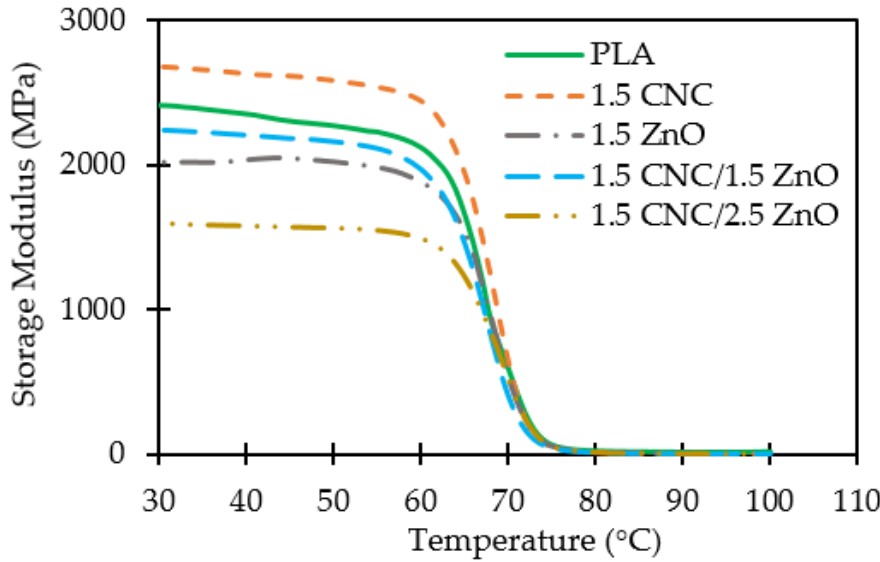

**Figure 4.** Representative curve of storage modulus vs. temperature PLA and corresponding nanocomposites.

The tanδ as the ratio of the loss modulus to storage modulus indicates damping characteristics of polymers and offers information of the mechanical energy dissipated as heat when force is removed in a sinusoidal pattern. A low tanδ intensity indicates that the material is a more elastic solid while a high tanδ intensity shows that the material is more viscous (Figure 5) [16]. All nanocomposite samples indicated a more elastic response (lower tanδ values) than pure PLA in the temperature range tested in this work. The lower tanδ corresponded to the nanocomposites reinforced with 1.5% CNCs, indicating the confinement effect of CNCs, and lower chain mobility of the PLA composite. However, as the content of ZnO increased, the nanocomposites showed more viscous behavior compared to PLA.

The glass transition temperature was also obtained from the position of the tanδ peak value on Figure 5, and there was no significant change in glass transition temperature of different formulations in comparison with pure PLA. Among different composites, the formulations that contain ZnO had lower $T_g$, suggesting the polymer chain deterioration. It was reported that the incorporation of ZnO nanoparticles into PLA can result in the polymer chain scission and reduce the molecular weight [10]. Nanocomposite samples with higher ZnO concentrations exhibited a higher peak value in the tanδ curve and lower storage modulus, suggesting the higher viscoelastic behavior. The diminishing tanδ value

with the addition of nanoparticles can be attributed to restrictions imposed by nanoparticles against the molecular motion of polymer chains. The decrement in tanδ is in agreement with a previous study [19].

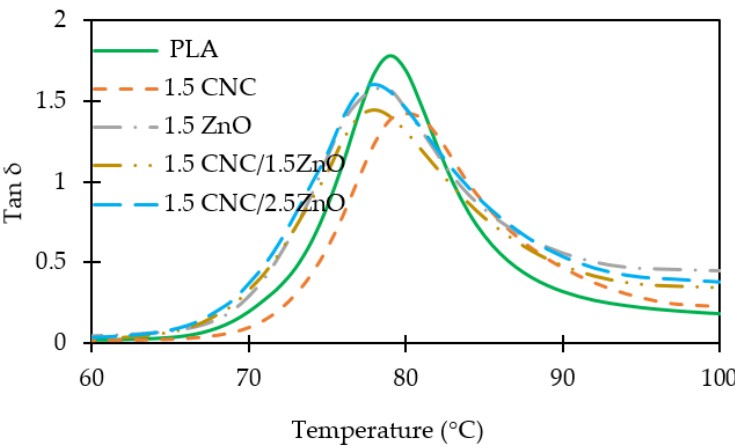

**Figure 5.** Representative curve of tanδ as a function of temperature for PLA nanocomposites reinforced with CNC and ZnO.

### 3.4. Scanning Electron Microscopy (SEM)

As observed in the SEM micrographs, Figure 6, a nonuniform dispersion of nano-ZnO may contribute to lower mechanical properties as the stress transfer is negatively impacted by the large particles. However, the presence of CNCs in nanocomposites with ZnO minimized the deteriorating effect of ZnO on viscoelastic properties of nanocomposites. In general, the combination of CNCs and ZnO nanoparticles improved the elastic response of nanocomposites, particularly for formulation 1.5CNC/1.5ZnO, which appeared to be the best formulation in terms of mechanical performance among nanocomposites containing ZnO nanoparticles.

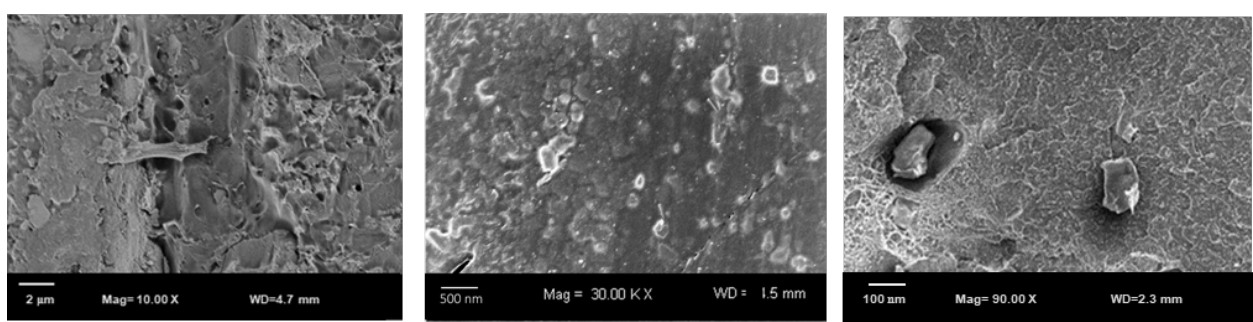

**Figure 6.** SEM cross-sectional images of PLA composites reinforced with 1.5% CNC and 1.5% nano-zinc oxide at different magnifications showing irregular distribution of zinc oxide particles.

### 3.5. Thermogravimetric Analyses (TGA)

The effect of ZnO nanoparticles and CNCs on the thermal properties of the PLA matrix under a nitrogen and oxygen atmosphere was examined using thermogravimetric (TGA) measurement, and Figures 7 and 8 illustrate the curves of weight loss against elevated temperature for PLA and its corresponding nanocomposites. The TGA curves indicate a single-step degradation for all the samples. The thermal stabilities can be analyzed by comparing the $T_{onset}$, $T_{50}$, and $T_{endset}$ values. $T_{onset}$ represents the initial decomposition temperature in which the nanocomposites start degrading, $T_{50}$ corresponds to 50% weight loss, and $T_{endset}$ is the end of the degradation temperature.

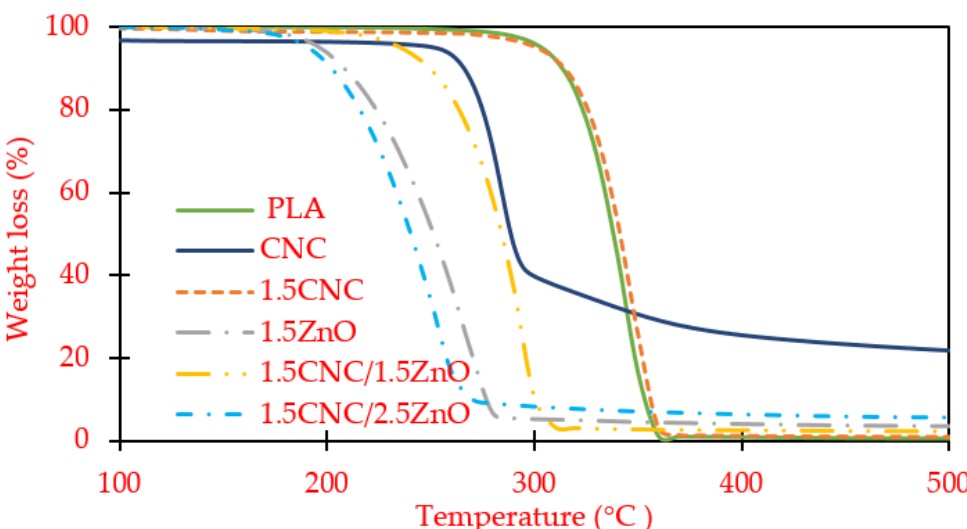

**Figure 7.** Percentage weight loss vs. temperature for each formulation.

From Table 3, the 1.5CNC nanocomposite showed a higher thermal stability than pure PLA as it exhibited a higher $T_{onset}$ in comparison with pure PLA. The increase in thermal stability can be attributed to the increase in PLA crystallinity by means of adding CNC, which can act as nucleating agents [20]. On the other hand, it can be seen that the presence of ZnO nanoparticles induced a shift of maximum degradation to lower temperature in comparison with pure PLA. This observation might be due to the catalytic role of ZnO nanoparticles at high temperature in the promotion and formation of lactide and related oligomers during PLA degradation.

**Table 3.** Thermal properties of each formulation from TGA.

| Sample | $T_{onset}$ (°C) | $T_{50}$ (°C) | $T_{endset}$ (°C) |
|---|---|---|---|
| PLA | 310.6 ± 5.6 | 338.4 ± 4. 3 | 355.6 ± 5.4 |
| 1.5CNC | 315.7 ± 9.0 | 341.6 ± 5.6 | 358.7 ± 6.3 |
| 1.5ZnO | 202.6 ± 7.4 | 251.6 ± 6.3 | 279.1 ± 7.8 |
| 1.5CNC/1.5ZnO [a*] | 247.2 ± 6.4 | 284.6 ± 8.4 | 302.8 ± 7.4 |
| 1.5CNC/1.5ZnO [b] | 210.1 ± 8.2 | 275.1 ± 7.8 | 295.9 ± 8.1 |
| 1. 5CNC/2.5ZnO | 195.9 ± 8. 9 | 240.6 ± 9.2 | 264.3 ± 9.5 |

*a and b represent thermo-degradation in nitrogen and in oxygen (70% oxygen, 30% nitrogen) environments, respectively.

Furthermore, at 50% weight loss, 1.5CNC/1.5ZnO possessed a significantly higher thermal stability than 1.5CNC/2.5ZnO. This indicates that the increasing loading of ZnO decreased the thermal stability of the composites as a result of further degradation at higher temperature. The thermo-oxidative degradation (70% $O_2$, 30% N atmosphere) of the composite with 1.5%CNC.1.5%ZnO exhibited a lower $T_{onset}$ temperature with a peak temperature of 280 °C. The lower oxidative temperature agrees with previous reported work [21]. Interestingly, the degradation process was not a well-defined single decomposition, with a second peak at 450 °C in the carbonization region. This can be attributed to residual char formation as observed in a previous study, which probably contributes to improve the flame retardancy of composites as noticed in burn tests [22]. Overall, the composites with the same ZnO content (1.5ZnO and 1.5CNC/1.5ZnO) formulation containing CNCs exhibited significantly higher thermal stability, suggesting the synergistic action and formation of a relatively strong network between composite components resulting in the shielding effect of CNC-ZnO due to char formation.

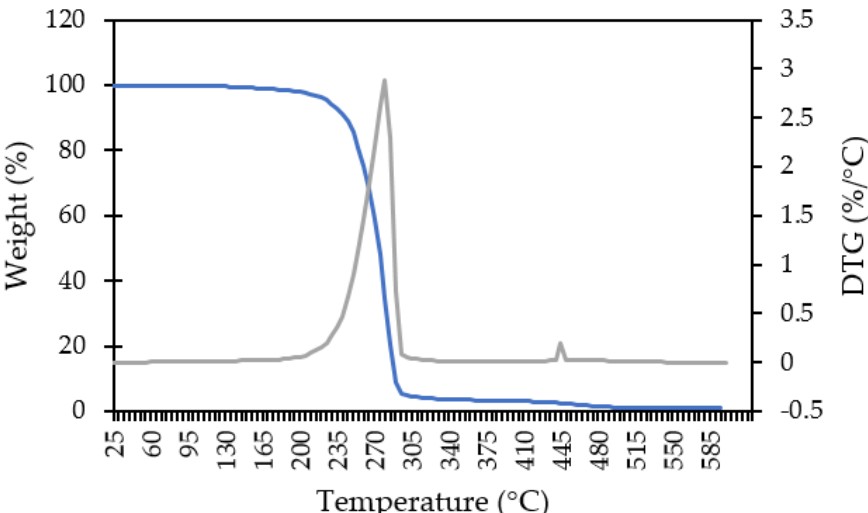

**Figure 8.** Thermogravimetric analysis of PLA composite with 1.5% CNC and 1.5% zinc oxide under oxidative degradation (70% air, 30% nitrogen).

### 3.6. Melt Flow Index (MFI)

The melt flow index values of virgin PLA resin and PLA nanocomposite reinforced with 1.5% CNC was 6.75 and 5.31 g/10 min, respectively. The noticeable decline in the MFI of CNCs-filled PLA nanocomposites could be traced to the flow burden of CNCs [23]. However, the composite formulation containing either ZnO nanoparticles or CNCs/ZnO nanohybrids exhibited a significant increase in the MFI. It seemed that ZnO nanoparticles attacked the main chain of PLA and reduced the molecular weight of PLA, and this, in turn, resulted in a high flow rate (more than 50 g/10 min), which was not measurable. This correlates with the results from TGA and DMA that ZnO nanoparticles cause intensive degradation to the polymer matrix [10].

### 3.7. Horizontal Burn Test

The horizontal burn test was conducted, as shown in Figure 9, to evaluate flame spread on nanocomposite samples, and the mean linear burning rate, type of burning, and mean mass loss are tabulated in Table 4.

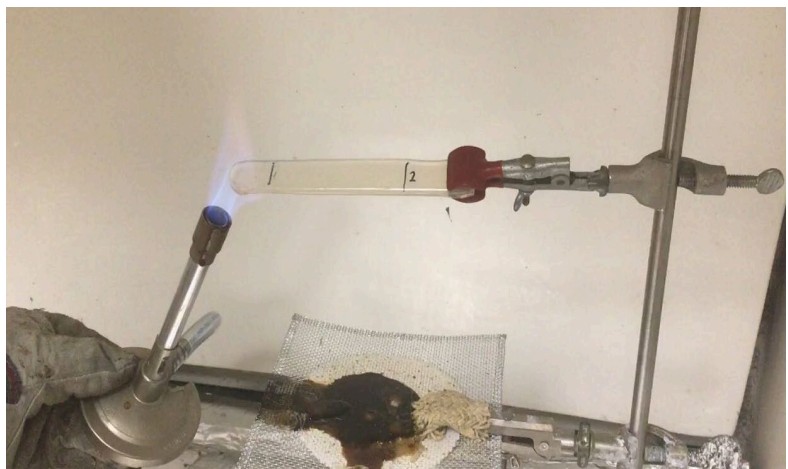

**Figure 9.** A picture showing the horizontal burn test (following ASTM D635–14 standard).

The addition of CNCs into PLA resulted in the nanocomposites with slightly higher flammability; however, the mass loss was drastically lower than what was observed for the pure PLA. The lower mass-loss can be attributed to the char formation and shielding

effect due to CNC and ZnO, respectively. Burn tests further bolstered this claim as the flame spread rate decreased with time. The continuous burning of 1.5CNC composites indicate that CNC can increase the strength of PLA against mass loss but perform poorly on flame spread rate. Fire retardancy is achieved when a material is made less prone to ignition and its combustion efficiency is decreased [21]. The high flammability of PLA was abated by the addition of ZnO, which resulted in noncontinuous burning. Again, this phenomenon might be due to the shielding effect of ZnO and a partial char layer of CNC that serves as an insulator on the outer surface of the test specimen. This shielding effect of ZnO diminishes the oxygen supply for combustion and ultimately leads to noncontinuous burning. The lower mass loss in CNC-ZnO-reinforced composites suggested a self-extinguishing property of nanocomposites probably owing to the synergistic effect due to shielding and char formation on the surface of nanocomposites.

**Table 4.** Flame retardancy observed for PLA and composites from horizontal burning test.

| Sample | Mean Linear Burning Rate (mm/min) | Type of Burning | Mass Loss (%) |
|---|---|---|---|
| PLA | 25.42 | Continuous | 69.1 |
| 1.5CNC | 30.20 | Continuous | 9.2 |
| 1.5ZnO | - | Noncontinuous | 3.3 |
| 1.5CNC/1.5ZnO | - | Noncontinuous | 3.2 |
| 1.5CNC/2.5ZnO | - | Noncontinuous | 4.0 |

### 4. Conclusions

The incorporation of nano-ZnO and CNC into the polymer matrix variably improved the mechanical and fire resistance of the PLA matrix. The study verified that ZnO nanoparticles can exhibit fire retardant properties. From the horizontal burning test, the nanocomposites with 1.5% CNC and 1.5% ZnO performed best in terms of having the least percentage of mass loss. Oxidative combustion did not show a well-defined single decomposition, denoting two different decomposition processes possibly due to strong char formation. In addition to the application of CNCs, the shielding effect from ZnO serves as an insulator and results in noncontinuous burning, which increases the fire retardancy of nanocomposites. The addition of cellulose nanocrystals led to a significant increase in the storage modulus of nanocomposites. By contrast, the addition of ZnO nanoparticles leads to a decrease in storage modulus, increase in tanδ, and lowering of the initial decomposition temperature. To overcome these deficiencies, further research is needed to understand the behavior of ZnO-overlaid CNCs in the PLA matrix and the interfacial interactions between the nanoparticles and polymer chains. Methylene dichloride, a safe polar geminal organic solvent, can be used to replace chloroform for manufacturing masterbatch.

**Author Contributions:** Conceptualization, D.S.B and S.G.B.; data curation, D.S.B.; investigation J.D.L and J.S.; methodology and writing, J.D.L and J.S.; writing reviewing and editing D.S.B, J.S. All authors have read and agreed to the published version of the manuscript.

**Funding:** Authors would like to acknowledge USDA-AFRI NIFA (Grant No. 2017–67022–26609) for their support to conduct this research project.

**Conflicts of Interest:** The authors declare no conflict of interest.

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
