# Peer review of "Role of Hybrid Nano-Zinc Oxide and Cellulose Nanocrystals on the Mechanical, Thermal, and Flammability Properties of Poly (Lactic Acid) Polymer"

_jcs, doi:10.3390/jcs5020043_

Round 1

Reviewer 1 Report

The work presents an interesting research topic, it is worth publishing after it is completed and corrected.

Line 172: 2.8 Fourier transform infra- please move it to the next line and correct the formatting

Line 176: wavelenght:  in FTIR spectra there is no wavelength only wavenumber, please correct it

 Paragraph 3.3 DMA

The author introduces abbreviations that he does not use later - please discuss it.

In Table 2, please explain the curves from which the individual figures (glass ransition temperature) are determined and add the values of the loss modulus.

Please correct the unit of Mpa to MPa. Please refer to the max tan delta values (are they correct? Please compare with the values given by the authors of other papers).

Line 245, line 276 and line 338: The authors explain some of the results by explaining this by changes in molecular weights and by referring to information provided elsewhere. Please determine the molecular weights of your compositions and you can then interpret it correctly.

Lines 278-280: Please provide literature sources confirming the noticed conclusions of the authors.

About tan delta: Based on the tan delta curves, please compare the homogeneity of the obtained compositions and compare them with the results of other research methods used in your work. This is a very important parameter.

Lines 306-307:  Please explain the reason for this.

The temperature values in DMA and TGA should be rounded to full values - the measurement error of the apparatus is greater than the accuracy with which the authors provided the values of individual temperatures.

Please format the references according to the journal template.

Major revision

Sincerely

Reviewer

Author Response

The authors would like to thank the reviewer for their time and effort involved in reviewing the manuscript thoroughly and providing valuable comments and suggestions. 

We have revised the manuscript per reviewer recommendations and suggestion. 

Reviewer 2 Report

This article is well done, which is why I agree with its publication in this form.

Author Response

The authors would like to thank and the reviewer for their time and effort involved in reviewing the manuscript. 

Reviewer 3 Report

In the reviewed article, the authors presented the use of a hybrid CNC and ZnO filler to modify PLA in order to improve mechanical properties, thermal and flame resistance properties. The authors prepared a hybrid filler, which was introduced into the matrix of the biopolymer in the form of a PLA masterbatch by extrusion.
The article requires corrections and additions. My comments:
In my opinion, Fig. 1 (line 97) confuses the microscopic photos of the CNC and ZnO filler.
The molar mass of the reagent used shows that it is zinc acetate dihydrate and not dehydrate as in the text.
What was the concentration of PLA in chloroform? Have the authors considered secondary re-agglomeration of the filler when drying the composite film?
During the extrusion of composites, raw materials were not dried, why? Processing of PLA materials requires the removal of any traces of moisture that causes its degradation.
Table 1 shows incorrect values ​​for the 1.5CNC / 1.5ZnO masterbatch and the 1.5CNC / 2.5ZnO composite.
Did the process of hybrid preparation change the size of the ZnO and CNC filler particles used?
Fillers and the hybrid should also be considered in FTIR tests.
Please mark the discussed characteristic bands in the FTIR spectra (Fig. 3).
TGA tests should take into account the measurements of the fillers and hybrids used.
Thermooxidative studies should also be carried out for fillers and on this basis, further conclusions should be drawn.
What is the mechanism of the degradative effect of ZnO on PLA macromolecules? Suggests excluding the influence of residual solvents and moisture.

Author Response

The authors would like to thank and the reviewer for their time and effort involved in reviewing the manuscript thoroughly and providing valuable comments and suggestions. 

We have revised the manuscript per your suggestions and comments.

Please note all the revisions are marked in red font color in the manuscript.

Round 2

Reviewer 1 Report

Paper is OK.

Reviewer 3 Report

The authors made significant changes and corrections of the reviewed work, which significantly increase the scientific value of the article. In my opinion, the revised work is suitable for publication in a J. Compos. sci. I leave the final decision to the editor.